# Image Security Based on Three-Dimensional Chaotic System and Random Dynamic Selection

**DOI:** 10.3390/e24070958

**Published:** 2022-07-10

**Authors:** Bo Ran, Tianshuo Zhang, Lihong Wang, Sheng Liu, Xiaoyi Zhou

**Affiliations:** School of Cyberspace Security, Hainan University, Haikou 570100, China; borandiligent@163.com (B.R.); zhang.tianshuo@163.com (T.Z.); WRwanglihong@outlook.com (L.W.); shengliu681@gmail.com (S.L.)

**Keywords:** chaotic system, image encryption, random dynamic selection

## Abstract

Image encryption based on a chaos system can effectively protect the privacy of digital images. It is said that a 3D chaotic system has a larger parameter range, better unpredictability and more complex behavior compared to low-dimension chaotic systems. Motivated by this fact, we propose a new image cryptosystem that makes use of a 3D chaotic system. There are three main steps in our scheme. In the first step, the chaotic system uses the hash value of the plaintext image to generate three sequences. In step two, one of the sequences is used to dynamically select confusion and diffusion methods, where confusion and diffusion have three algorithms, respectively, and will produce 3^2n^ (*n* > 100) combinations for encryption. In step three, the image is divided into hundreds of overlapping subblocks, along with the other two sequences, and each block is encrypted in the confusion and diffusion process. Information entropy, NPCR, UACI results and various security analysis results show that the algorithm has a better security performance than existing, similar algorithms, and can better resist clipping, noise, statistical analysis and other attacks.

## 1. Introduction

Digital imagery is a vital medium for the presentation of information. Digital images break down language barriers and promote ideological exchange. They have been widely used in medical, military, financial, judicial and other fields. They also face the problem of information leakage, and some even involve national security, such as pictures taken by military satellites, military facility maps, building maps of financial institutions, etc.

Generally, there are two ways to protect the privacy of digital images: image watermarking and image encryption. The former mainly refers to the method of hiding digital watermarking in plaintext images to protect the copyright of the plaintext images and prevent illegal transmission. The latter refers to the operation of the pixel position and value of the image through specific reversible mathematical transformation, so that the plaintext images and the corresponding ciphertext images are as uncorrelated as possible, to prevent unauthorized third parties from learning the specific content information of the plaintext image.

Unfortunately, traditional encryption algorithms such as DES, AES [1,2] are not ideal for real-time image encryption, as they do not fully consider image data’s large capacity, strong pixel correlation and high redundancy. For example, in electronic code books (EBCs), the ciphertext image generated by AES encryption still has an obvious plaintext image outline. In 1997, researchers found that chaotic systems have initial sensitivity, pseudo randomness, ergodicity, parameter sensitivity and unpredictability, which could be combined with the characteristics of images. Chaotic systems are widely used in image encryption [3,4,5,6,7,8,9,10,11,12,13,14,15,16,17,18,19,20]. Frequently used chaotic systems include Logistic [21], Sine [22] and Tent [23]. 

Compared with classic chaotic systems, which are one-dimensional, hyperchaotic systems have better randomness, ergodicity and a wider parameter range, resulting in the generation of more chaotic sequences. Therefore, Zheng et al. [17] presented a novel form of image encryption by combining dynamic DNA sequence encryption and the improved 2D logistic sine map, which improves the small parameter space problem of one-dimensional chaotic systems. Gao et al. [24] designed a 2D chaotic system by combining two one-dimensional chaotic systems. The experimental results showed that the two-dimensional hyperchaotic map had more complex dynamic characteristics and randomness. Hua et al. [25] proposed an image scheme with 2D logistic-adjusted-Sine chaotic systems. The performance evaluation showed that it had better ergodicity and unpredictability than many existing chaotic systems. However, the trajectory of these chaotic systems does not distribute in the entire phase space, which means that they lack complex dynamic behavior and are vulnerable to interruption [26,27]. Research shows that the existing encryption algorithms for chaotic systems with a small binding range, simple random behavior and low ergodicity may be attacked [18,28,29].

Pseudo-random numbers generated by chaotic systems are used to scramble and diffuse links in the encryption process to obtain cryptographic images with a good encryption performance. The variation in pixel positions or values may be deduced based on a single permutation or diffusion method. Wang et al. [30] proposed a hybrid encryption algorithm of pixel values, pixel bits and binary bits. This introduced both linear and nonlinear diffusion operations and mixed the permutation and diffusion procedures in one step, which may prevent attackers from knowing which encryption method is being used. Wang et al. [31] utilized multiple one-dimensional chaotic systems for block encryption to mix the characteristics of multiple chaotic systems, making them more difficult for attackers to crack. Xian et al. [32] proposed a form of image encryption based on chaotic sub-block permutation and digital selection diffusion. One of several methods is selected to execute the diffusion, forcing a cracker to crack each algorithm to successfully obtain a clear text image. Khan et al. [33] introduced an efficient block image encryption, relying on the dynamic selection expansion of the correlation coefficient of each block. Each block image is encrypted with block-related parameters, which expands the variables and makes cracking more difficult. Yavuz et al. [5] introduced a switching chaotic image encryption algorithm based on the content-sensitive dynamic function. The continuous switching of functions during encryption gives the final image different characteristics of function encryption. Generally, these algorithms dynamically choose encryption methods. The method used in the above paper selects one of various encryption methods or constantly changes parameters during encryption to hide the change laws of pixel value and position. However, due to the limitations regarding the number of encryption methods and the small parameter range, the correct encryption methods and parameters can be found through exhaustive methods.

In this paper, an image encryption scheme named 3D-PHSL, combining a hyperchaotic system and random dynamic encryption, is proposed. If the same chaotic sequence is used for three shuffling and three diffusion operations, the effects of shuffling and diffusion may be offset. Therefore, we designed a three-dimensional, chaotic system to ensure that the chaotic sequences are not repeated. To expand the range of parameters and generate more chaotic sequences, a 3D chaotic system is designed by combining polynomial, Logistic, Sine and Hermon chaotic systems. This makes up for the shortcomings of their chaotic behaviors, such as their not being complex enough, being insensitive to initial conditions, and not having a large enough parameter interval [17,24,25]. The image is divided into hundreds of subblocks; each is encrypted by one of three confusion methods and one of three diffusion methods. Meanwhile, three random sequences are generated by the chaos system. One is used to select permutation and diffusion methods; the other two are used for block encryption.

The contributions of this paper are as follows:
A three-dimensional hyperchaotic system with wide parameter range and good randomness is designed.The block-divided methods make the image blocks overlap, which enables the algorithm to become more sensitive to plaintext. The pixel values in each image block are encrypted by different permutation and diffusion methods.The single permutation-diffusion process of the traditional methods is optimized. It uses 9^n^ (*n* > 100) combinations of encryption algorithms to obtain the final encrypted image.


As a result, the security of the system proposed in this paper is more dependent on the key. Even if individual permutation and diffusion algorithms are cracked by an attacker, it is difficult to recover the ordinary image from the encrypted image. Experimental results show that the proposed algorithm can effectively resist differential attacks, cropping attacks, noise attacks, etc. The performance evaluations, including ergodic graph, bifurcation diagram, Lyapunov exponent spectrum, initial value sensitivity, information entropy and adjacent pixel correlation, show that 3D-PHSL produces more chaotic sequences than the existing chaotic ones. Furthermore, it has better ergodicity, more parameters and a wider parameter range, which guarantees that it has better unpredictability and stability. 

The rest of this paper is organized as follows: Section 2 introduces the state-of-the-art of chaotic systems and proposes the novel chaotic system 3D-PHSL, as well as introducing the advantages of the new system. Section 3 further presents a random dynamic encryption based on 3D-PHSL and its corresponding decryption scheme. Section 4 analyzes the security performance, including a statistical attack analysis, differential attack analysis, detailed attack analysis, anti-noise attack and computational complexity. Section 5 analyses the experiment results of the proposed algorithm in color images. Section 6 discusses some future research based on the conclusions of the paper.

## 2. Chaotic System

### 2.1. Definitions

The definition of the Logistic chaotic system is shown in Equation (1): (1)$${\;x}_{n+1}={\mu x}_{n}\left(1-x_{n}\right)$$
when the value of μ ϵ [3.57, 4], the system is in a chaotic state.

The definition of Sine chaotic system is shown in Equation (2):(2)$$x_{n+1}=\mu \sin\left({\pi x}_{n}\right)$$
where μ is the parameter of the chaotic system. When μ ϵ [0.87, 1], the system is in a chaotic state.

The definition of the Hénon chaotic system is shown in Equation (3):(3)$$\{\begin{array}{c} x_{n+1}=1-ax_{n}^{2}+y_{n}\\ {\;y}_{n+1}=bx_{n}\; \end{array}$$
where a and b are the control parameters of the chaotic system. When b = 0.3 and a ϵ [1.06, 1.22]∪[1.27, 1.29]∪[1.31, 1.42], the system is chaotic. The bifurcation diagrams of Logistic, Sine and Hénon are shown in Figure 1a–c. The trajectory is simple, the parameter range is small, and the low-dimensional chaotic system cannot meet the needs of encryption. Therefore, a hybrid and unpredict 3D chaotic system is proposed. Figure 1d is a bifurcation of a 3D chaotic system. Compared to Figure 1a–c, Figure 1d is nearly distributed throughout the space. The definition is as shown in Equation (4)
(4)$$\{\begin{array}{c} x_{n+1}=ay_{n}\left(1-x_{n}^{2}+z_{n}\right)\text{mod}\;1\\ y_{n+1}=bsin\left(\pi z_{n}\left(1-y_{n}\right)\right)\text{mod}\;1\\ z_{n+1}=x_{n}^{3}+y_{n}^{2}+z_{n}+c\;\text{mod}\;1\; \end{array}$$
where a, b and c are the control parameters of the chaotic system. When a, b, c ϵ (−∞, 0) ∪ (0, +∞), the system is chaotic.

### 2.2. Performance Analysis of PHSL Chaotic System 

A good chaotic system produces sequences with good randomness and maintains the sensitivity of the initial values. It can be evaluated by Lyapunov exponent and permutation entropy, etc.

#### 2.2.1. Initial Value Sensitivity Analysis

The initial sensitivity test aims to evaluate the sensitivity of the chaotic system. If there is a small change in the initial value, the generated sequence will change greatly. The initial values of the first group are x = 78.66666, y = 56, z = 67, respectively. The initial values of the second group are x = 78.66667, y = 56, z = 67. Two groups of initial values with little difference are input into the chaotic system and iterated 50 times to generate x1 and x2 chaotic sequences. The difference between the two groups is 0.00001. It can be seen from Figure 2 that the chaotic sequences greatly differ when the initial value difference is tiny. Therefore, the chaotic system is extremely sensitive to the initial values.

#### 2.2.2. Bifurcation Diagram Analysis

The bifurcation diagram shows the randomness of the chaotic system. The more complex the trajectory, the better the randomness. From Figure 1a–d, we can see that the bifurcation diagrams of Logistic, Sine and Hénon are simpler and have a smaller range than our proposed chaotic system. This shows that the proposed system has better randomness and complexity than the classical chaotic system.

#### 2.2.3. Lyapunov Exponent Analysis

The Lyaponuv exponent represents the numerical characteristics of the average exponential divergence of adjacent tracks in phase space, and is one of the characteristics used to identify numerical values of chaotic motion. The convergence or divergence velocity of a system trajectory can be denoted by λ, the Lyapunov exponent (LE) [34]. A positive LE indicates that even small changes in the initial state will result in a completely different output. Therefore, the dynamic mapping is chaotic when λ > 0. From Figure 3d, it can be seen that the maximum LE of the proposed chaotic system is greater than 0. Compared with the Lyapunov exponent values of the three chaotic systems in Figure 3a–c, 3D-PHSL has a wider chaotic range and a larger LE. A 3D stereogram of LE is shown in Figure 3e–g. The values of the two surfaces are greater than 0. The value in another surface is less than 0 only when it is close to 0, and the rest of the cases are also greater than 0. Therefore, the 3D-PHSL system has better randomness and ergocity. 

#### 2.2.4. Permutation Entropy

Permutation Entropy (PE) [35] is an index used to measure the complexity of a sequence. It is used to evaluate the complexity of time series. The more regular the sequence, the smaller the permutation entropy; the more complex the sequence, the greater the corresponding permutation entropy. It detects the dynamic changes in a time series by comparing the values of adjacent time series. The higher the PE of the sequence produced by the chaotic system, the better the pseudo-randomness of the sequence. 

The following is the calculation process of PE:


*Sequence:*

X={ x(1), x(2)......x(n)}




*Step 1: reconstruct the X sequence.*



*Reconstruction matrix:*

$$\left[\begin{array}{c} x\left(1\right)\;x\left(1+t\right)\;\cdots \;x\left(1+\left(m-1\right)t\right)\\ \vdots \;\vdots \;\vdots \;\\ x\left(j\right)\;x\left(j+t\right)\;\cdots \;x\left(j+\left(m-1\right)t\right)\\ \;\vdots \;\vdots \;\vdots \;\\ \;x\left(K\right)\;x\left(K+j\right)\;\cdots \;x\left(K+\left(m-1\right)t\right) \end{array}\right]$$



t *is the delay time and* m *is the embedding dimension.* j *= 1,2,3...* K.

*Step 2: Calculate the sequence*(Si(i = 1..k) *corresponding to each line component of the reconstruction matrix and the corresponding probability*(Pi(i = 1...k)).

*Calculation method of sequence: each row of the reconstructed matrix is sorted in ascending order according to the value, and their corresponding index values will form a sequence* S_i_(1 × m). *From this, we can calculate that there are* m! *permutations in this sequence.*


*Calculate probability of* S_i_*: Count the number of occurrences of* S_i_
*in all permutations, and calculate the probability* P_i_.


*Step 3: Calculate the Permutation Entropy.*

$$PE=-\overset{K}{\underset{i=1}{\sum }}P_{i}lnP_{i}\;$$




*
**Normalization processing:**
*

PE=PE/lnm!



Figure 4 shows that the entropy of the arrangement of chaotic sequences produced under different parameters is greater than 0.98, so the sequences produced by the proposed systems tend to be more random. As shown in Table 1, the PE of multidimensional chaotic systems is represented by the average of multiple chaotic sequences. It can be seen that the PE value proposed in this paper is the largest, and therefore has the best randomness compared with [21,22,36,37,38,39].

#### 2.2.5. NIST Test 

The potential of the proposed is investigated using the test developed by the National Institute of Standards and Technology (NIST), since it is the authoritative statistical test essential in quantifying the randomness level that exists in the stream sequence generated through 3D PHSL. This NIST-based test suite comprises 15 tests in which a collection of *p* values is generated during the enforcement of each of the tests corresponding to the set of stream sequences. When *p* ≥ 0.01 in a test, we consider that the sequence has passed the test. We divided the random sequence generated by 3D PHSL into 200 groups for testing. The success rate of each NIST test is shown in Table 2. Finally, the rate of random sequence passing the test is close to 100%.

## 3. Encryption Procedure

An image is divided into hundreds of subblocks, then three sequences are generated according to the proposed chaotic system. One of the sequences involves the selection of the permutation and diffusion methods, and other two are used for encryption. Encrypted images with good security can be obtained after only one round of encryption. The encryption process can be described as follows:

Step 1:

The plaintext image of size of m×n is converted into 512-bit hash value h by SHA-512, and then, h is divided into 16 parts to obtain Ki (i = 1, 2, 3... 16). The initial value and parameters (r_1_, r_2_, r_3_, a, b, c) are obtained according to Equation (5). The length of each initial value and parameter is 32 bits. Thus, the key consisting of r_1_, r_2_, r_3_, a, b and c is generated. Input the initial values and parameters into the chaotic system, iterate 1000 times and then iterate m × n times to obtain the chaotic sequences H, I and O (m × n), and quantify the O sequence to the range of [0, 2].
(5)$$\begin{array}{l} r_{1}=K_{1}⊕K_{16}⊕K_{12}\\ r_{2}=K_{3}⊕K_{14}⊕K_{10}\\ r_{3}=K_{5}⊕K_{8}⊕K_{15}\\ a=K_{7}⊕K_{13}⊕K_{6}\\ b=K_{9}⊕K_{4}⊕K_{1}\\ c=K_{11}⊕K_{5}⊕K_{2} \end{array}$$

Step 2:

The image block formed by every two rows of the image is scrambled according to the value of H and I. Sequence O is used to randomly select which permutation and diffusion algorithm will be used to process image blocks. The selection of the permutation algorithm is achieved by Equation (6). After the permutation of each image block, P1 is obtained.
(6)$$\{\begin{array}{c} Permutation\;algorithm\;1,\;O\left(i\right)=0\\ Permutation\;algorithm\;2,\;O\left(i\right)=1\\ Permutation\;algorithm\;3,\;O\left(i\right)=2 \end{array}$$

Step 3:

The image block formed by every two columns of the image (P1) is scrambled according to the value of H and I. The selection process is shown in Equation (6). After the above operations, the pixels in one position can be replaced anywhere in the image. Two chaotic sequences (H and I) need to be used in the algorithm. P2 is obtained after scrambling each image block.

The three permutation algorithms (Algorithms 1–3) used in this research are as follows:

The S and M sequences used by the three permutation algorithms are intercepted from sequences H and I. K is the representation of the output image block in the following specific encryption algorithm.
**Algorithm 1** Permutation Input:Plain image blocks P, sequences S (S is half the size of P)Output: Scrambled Image Block K1: K ← [];2: K ← [S P];   %Combine S and P into one matrix3: K ← sortrows(K,1);    %Move rows in ascending order according to the first column4: K ← K(:, 2:3);5: Output Scrambled Image Block K

**Algorithm 2** Permutation
Input:Plain image blocks P, sequences S, M (S and M are half the size of P)Output: Scrambled Image Block K1: Q, X ← sort (S), sort (M);   %Ascending order2: f = length(S);3: for i = 1 to f do  %Quantify the elements in S to [1,f] 4:  for j = 1 to f do5:   if Q(i) = = S(j) then6:    S(j)←i;7:   end8:  end 9: end 10: for i = 1 to f do   Quantify the elements in M to [1,f] 11:  for j = 1 to f do12:   if X(i) = = M(j) then13:    M(j)←i;14:   end15:  end16: end 17: for i = 1 to f do   %Scramble two columns of P according to the quantized elements in S and M18:   K(i, 1)←P(S(i), 1);19:   K(i, 2)←P(M(i), 2);20: end 21: Output Scrambled Image Block K

**Algorithm 3** Permutation Input:Plain image blocks P, sequences S, M (The size of S is f and the size of M is N.)Output: Scrambled Image Block K1: [F, N]←size(P);   % Image Block Size2: for i = 1 to F do    %The distance to shift each row is determined by the value of the S sequence.3:   K(i, :)←circshift(P(i, :), S(i), 2);4: end 5: for i = 1 to N do   %The distance to shift each column is determined by the value of the M sequence.6:   K(:, i)←circshift(P(:, i), M(i), 1);7: end 8: Output Scrambled Image Block K

Step 4: 

Partition and diffuse P2. Firstly, the image block composed of every two rows of the image is diffused, and the diffusion algorithm is selected according to O. The selection process is achieved by Equation (7). Two chaotic sequences, H and I, are used in the diffusion process. C1 is obtained after diffusion operations on each image block.
(7)$$\{\begin{array}{c} Diffusion\;algorithm\;1,\;O\left(i\right)=0\\ Diffusion\;algorithm\;2,\;O\left(i\right)=1\\ Diffusion\;algorithm\;3,\;O\left(i\right)=2 \end{array}$$

Step 5: 

Take every two columns of C1 as image blocks, select the diffusion algorithm based on the corresponding value of the O and perform the diffusion operation on each image block again. The selection process is shown in Equation (7). C2 is obtained from each image block. The encryption process is shown in Figure 5. 

The image encryption algorithm we proposed is symmetric, so the decryption algorithm is the inverse process of the encryption algorithm. When decrypting, we need to obtain the same key as when encrypting. The key is split into r_1_, r_2_, r_3_, a, b and c, and these are input into 3D-PHSL to generate the same chaotic sequence as the encryption. After decryption, the plaintext image can be obtained according to the operation opposite the above encryption steps.

The three diffusion algorithms used in this paper are as follows:

Diffusion Algorithm 1:

Intercept the sequences (S, M) from H and I. The Equation is shown in Equation (8),
(8)C(i)=S(i)⊕M(i)⊕P(i) 

Diffusion Algorithm 2:

$$\{\begin{array}{c} \;f=length\left(S\right)\;\\ T\left(1\right)=\left(S\left(1\right)\times 10^{6}\right)\;mod\;256\;\\ T\left(2\right)=\left(M\left(1\right)\times 10^{6}\right)\;mod\;256\;\\ u=3.99+90.00001\times M\left(1\right)\;\\ x1\left(1\right)=0.001\times S\left(1\right)\;\\ x1\left(i+1\right)=u\times x1\left(i\right)\times \left(1-x1\left(i\right)\right)\left(i=1,2...f-1\right)\;\\ T\left(i+1\right)=(floor\left(T\left(i-1\right)+\left(T\left(i\right)/255\right)+x1\left(i-1\right)\times 10^{5}+K\left(i-1\right)\right)mod\;256\;\left(i=2,3...f+1\right)\; \end{array}\;$$
where S and M are the generated random sequence, and K  is the previous diffused image.

Diffusion Algorithm 3:

$$\{\begin{array}{c} f=length\left(S_{0}\right)\;\\ P1\left(i,1\right)=\left(P\left(i,1\right)+S\left(i,1\right)\right)mod\;256\;\left(i=1,2\right)\\ P1\left(i,t\right)=\left(P\left(i,t\right)+P1\left(i,t-1\right)+S\left(i,t\right)\right)mod\;256\;\left(i=1,2;t=2,3\ldots f/2\right)\\ P1=fliplr\left(P1\right)\;\\ T\left(i,1\right)=\left(P1\left(i,1\right)+S\left(i,1\right)\right)mod\;256\;\left(i=1,2\right)\;\\ T\left(i,t\right)=\left(P1\left(i,t\right)+T\left(i,t-1\right)+M\left(i,t\right)\right)mod\;256\;\left(i=1,2;t=2,3\ldots f/2\right) \end{array}$$
where *S*_0_ is the random sequence, *S* and *M* are the random matrix obtained by changing the size of the original random sequence from one row to two rows and halving the number of columns, and *K* is the image before diffusion. Figure 6 is an example of permutation and diffusion.

## 4. Experimental Results and Performance Analysis

In the experiments, a computer with the Mac OS operating system, 8 GB of memory, a 2.4 GHz central processing unit, and Matlab r2019b was used as the simulation system. The plain text images selected in this paper were Lena, plane, pepper, baboon, black and white. Experiments were carried out on images with sizes of 256 × 256 and 512 × 512, respectively.

### 4.1. Encryption and Decryption Results

Figure 7 shows the original plaintext image and the encryption and decryption results. The decrypted image is completely consistent with the original one. 

### 4.2. Security Analysis

In this section, the key sensitivity, histogram, correlation, NPCR, UACI, information entropy and time-complexity of the proposed algorithm are analyzed by their ability to protect against various attacks. 

#### 4.2.1. Key Security Analysis

Take the 512 × 512 Lake as an example. The initial values x, y and z of chaotic system were set as 0.68, 0.83 and 0.61, respectively. The three initial values were changed slightly for decryption. Figure 8c–e show the decryption results after the minor changes in x, y or z. As is shown, the image cannot be recovered, which indicates that the proposed algorithm has good key sensitivity. 

#### 4.2.2. Histogram Analysis

A histogram is used to describe the distribution of pixels in an image. Even if a pixel position is changed, the attacker can easily obtain the image information by analyzing the histogram. Therefore, whether the pixels in the histogram are evenly distributed is also an important factor when measuring the encryption performance. Figure 9 shows the histogram of the image before and after encryption. The histogram pixels of the ciphertext image are evenly distributed, and no information related to the plaintext image can be obtained from the pixel distribution. This proves that the encryption algorithm has strong security. 

#### 4.2.3. Adjacent Pixel Correlation

An encryption algorithm needs to reduce the correlation of adjacent pixels to prevent attackers from obtaining useful information. Figure 10 shows the distribution of un-encrypted and encrypted Lena in horizontal, vertical and diagonal directions. As can be seen from Figure 10, the correlation in the three directions of the encrypted image is evenly distributed throughout the whole image, indicating that the proposed algorithm effectively reduces the correlation between adjacent pixels. The relevant calculation formula can be found in Equation (9).
(9)$$\left\{ \begin{array}{l} r_{xy}=\frac{\mathrm{cov}\left(x,y\right)}{\sqrt{D\left(x\right)}\sqrt{D\left(y\right)}};\\ \mathrm{cov}\left(x,y\right)=\frac{1}{N}\sum_{i=1}^{N}\left(x_{i}-E\left(x\right)\right)\left(y_{i}-E\left(y\right)\right);\\ D\left(x\right)=\frac{1}{N}\sum_{i+1}^{N}{\left(x_{i}-E\left(x\right)\right)}^{2};\\ E\left(x\right)=\frac{1}{x}\sum_{i=1}^{N}x_{i}. \end{array}\right.$$


Table 3 lists the correlation coefficients in the three directions of the encrypted image, and compares the correlation coefficients with those in [8,30,38,40,41]. Table 3 shows that the correlation coefficients of the proposed algorithm are lower. The optimal value of the proposed algorithm accounts for 41.7% in 12 tests, while those in the literature [42] account for 25%. Regarding the mean value of each test value, the literature [42] presents a value of 0.003333333, while our value was 0.002766667, which is 5.6% lower than that of [42]. This further shows that the correlation between pixels is eliminated with our proposed scheme.

#### 4.2.4. Anti-Differential Attack

The pixel change rate (NPCR) and unified average changing intensity (UACI) are two indicators that can be used to evaluate differential attacks. They can be used to describe the differences between two images. NPCR and UACI are defined as Equation (10):
(10)$$\left\{ \begin{array}{l} NPCR\left(C_{1},C_{2}\right)=\sum_{j=1}^{i=N}\sum_{i=1}^{i=M}\frac{D\left(i,j\right)}{M\times N}\times 100\%;\\ UACI=\left(C_{1},C_{2}\right)=\sum_{j=1}^{i=N}\sum_{i=1}^{i=M}\frac{\left|c_{1}\left(i,j\right)-c_{2}\left(i,j\right)\right|}{M\times N\times 255}\times 100\%, \end{array}\right.$$
where *C*_1_ and $C_{2}$, respectively, represent different encrypted images. *M* × *N* indicates the size of the image. If *C_i_*(*i*,*j*) ≠ *C*_2_(*i*,*j*), then D(i,j)=1; otherwise, D(i,j)=0. 

In this experiment, the UACI and NPCR of various images are calculated and compared with the other schemes. Table 4 shows that the NPCR of the proposed scheme is closer to the ideal values when compared with [8,40]. The optimal rate is 50%, while [42] only reached 33.3%. In Table 5, the optimal UACI value of the proposed scheme reached 66.7%, while [8.40] have a value of only 16.7%. The average values of NPCR and UACI of the proposed algorithm are also closer to the theoretical values than those in the other literature. For a highly sensitive encryption method, NPCR should be close to 99.6094% and UACI should be close to 33.4635% [43]. In conclusion, the UACI and NPCR of the proposed scheme are closer to the theoretical values.

#### 4.2.5. Information Entropy

Information entropy is important to measure the degree of information order, and its value is positively correlated with the degree of chaos in the system. Information entropy is defined as in Equation (11).
(11)$$H\left(s\right)=\sum_{i=0}^{2^{L}-1}p\left(s_{i}\right)\log_{2}\frac{1}{p\left(s_{i}\right)}$$
where $p\left(s_{i}\right)$ represents the probability of the occurrence of the symbol $s_{i}$. The closer the information entropy is to 8, the more disordered the image. Table 6 lists the information entropy of different test images and corresponding ciphertext images. The information entropy of different test images in the proposed algorithm is generally better than that of [8,23,38,40,41,42]. The optimal rate of the proposed method is 80%, while the optimal rate of [42] is only 60%. The probability of each value in the ciphertext image is almost the same, and the ciphertext information entropy obtained by the proposed algorithm is close to the ideal value of 8. This indicates that our scheme better resists differential attacks.

#### 4.2.6. Robustness Analysis

Figure 11 shows the results of the cropping attack and the recovery for the encrypted Lena at different positions and levels. Figure 12 shows the results of different degrees of noise interference and the restoration of encrypted Lena. The restored image can still be distinguished after the encrypted image is cropped by 70%, which proves that the proposed algorithm has a certain ability to resist attacks.

#### 4.2.7. Test Analysis

$\chi^{2}$ can be used to quantitatively describe the degree to which the image deviates from the absolute uniform distribution. This can be defined in Equation (12).
(12)$$\chi^{2}=\sum_{i=0}^{255}{\left(p_{i}-\bar{p}\right)}^{2}∕\bar{p}$$
where $\bar{p}$ is the average frequency of all pixels 
and *p_i_* is the 
frequency of pixels. When *χ*^2^ < 
290, the pixel distribution is uniform. Table 7 
shows that the *χ*^2^ value 
of the encrypted image is much lower than that of the plaintext image. The 
above results show that the proposed algorithm successfully unify the frequency 
of occurrence for pixel values in the encrypted image.

#### 4.2.8. Encryption Speed

In this section, we used a Mac OS with 8GB memory and 2.4GHz CPU and MATLAB r2019b to test the encryption and decryption time. If the image size is 256 × 256, the encryption time is 0.176182 s and the decryption time is 0.189482 s. If the size is 512 × 512, the encryption time is 0.526771 s and the decryption time is 0.514788 s. Some papers, such as [44,45,46], only list the execution time of their own algorithm; However, other papers such as [9,47,48] make comparisons with other algorithms in execution time, ignoring the different hardware facilities. For this reason, we refer to paper [44,45,46] and only list the execution time of our own algorithm. 

The experimental results show that the encryption and decryption times of the algorithm proposed in this paper are short, as shown in Table 8. 

#### 4.2.9. Key Space

The key in this paper is used to generate chaotic sequences. The process of key generation is described in step 1. The key space is determined according to the range of the initial value and parameters. Key space is (2^32^)^6^ = 2^192^, far greater than 2^100^, which can resist brute force attacks. The key generation process is in step 1.

#### 4.2.10. Time Complexity

m × n-size images were selected for the experiments. The time complexity of encryption algorithm mainly focuses on the random selection of the three confusion methods and the three diffusion methods. The average time complexity of the three permutation algorithms is O(m^2^) or O(n^2^), while that of the three diffusion algorithms is O(m) or O(n). Therefore, the time complexity of the whole image encryption algorithm is O(m^2^ + n^2^ + m + n). Table 9 shows that the time complexity of our algorithm is lower than that of other papers [49,50,51].

## 5. Application of Color Image

The experimental results for color image encryption and analysis prove that the algorithm is also suitable for color images, so it has wide application prospects.

### 5.1. Color Image Encryption and Decryption Results

The R, G and B Lena channels of 256 × 256 color are encrypted and decrypted by the proposed algorithms. The results are shown in Figure 13.

### 5.2. Test and Analysis of Color Encrypted Image

#### 5.2.1. Histogram Analysis

Histograms can directly reflect the distribution of pixels in the image. The pixels with different ciphertext image values need to be as evenly distributed as possible. When the histogram is unevenly distributed, the attacker can perform statistical attacks to obtain the information from the histogram. Figure 14 is a histogram of the R, G and B plaintext channels and the corresponding ciphertext. It can be seen that the histogram of ciphertext image is evenly distributed, which shows that the proposed algorithm is resistant to known cryptographic attacks such as statistical attacks.

#### 5.2.2. Correlation Analysis

Table 10 shows the horizontal correlation, vertical correlation and diagonal correlation values of R, G and B channels before and after encryption. After encryption, the image correlation is significantly decreased. Figure 15 shows the horizontal correlation of R, G and B channels before and after encryption. The image correlation is significantly decreased after encryption.

#### 5.2.3. Cropping and Decryption Results of Color Image

Encryption algorithms need to be able to resist a certain degree of cropping attacks. Figure 16 shows the ciphertext image corresponding to the plaintext image under the R, G, and B channels of the same image, and the decrypted image after different levels of cropping. The experimental results show that when the ciphertext image is cropped by 80%, the general outline of the plaintext image can still be seen from the decrypted image, which indicates that the proposed algorithm has sufficient security against cropping attacks.

## 6. Conclusions

In this paper, an image encryption scheme combining a hyperchaotic system and random dynamic encryption is proposed. To expand the range of parameters and generate more chaotic sequences, a 3D chaotic system is designed by combining polynomial, Logistic, Sine and Hermon chaotic systems. Lyapunov exponent, bifurcation diagram and initial value sensitivity tests prove that the parameter range of the chaotic system is wider, and the generated chaotic sequence is unpredictable and available. Random dynamic encryption is used to protect the security of the plaintext image. The final encrypted image will have a combination of 9^n^ (*n* > 100) algorithms, which is greatly improved compared with the existing schemes.

Through simulation and comparison, the security is verified from the aspects of statistical analysis attack, differential attack and noise attack. Analysis shows that the scheme has a large key space and can resist brute force attacks. The histogram of different images encrypted by the proposed scheme is almost the same, and the correlation coefficient is close to 0. The values of information entropy, NPCR and UACI are close to the ideal value, which proves that the scheme has a good encryption performance and high level of security.

However, there is still room for us to improve the efficiency. For example, confusion and diffusion are used in parallel or in combination to improve efficiency and reduce the time complexity of confusion and diffusion algorithms, to avoid attackers estimating the algorithms through the encryption time.

## Figures and Tables

**Figure 1 entropy-24-00958-f001:**
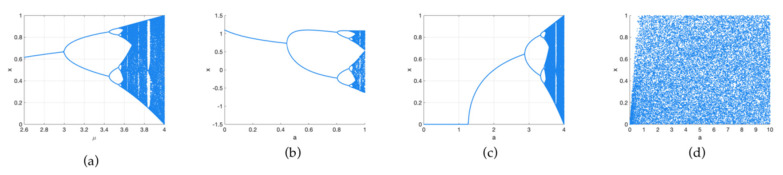
Bifurcation of different chaotic system: (**a**) bifurcation diagram of Logistic map; (**b**) bifurcation diagram of Hénon map; (**c**) bifurcation diagram of Sine map; (**d**) bifurcation diagram of proposed map.

**Figure 2 entropy-24-00958-f002:**
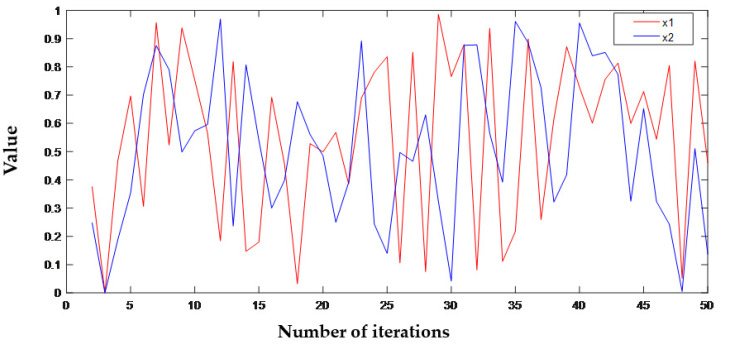
The effect of small initial changes.

**Figure 3 entropy-24-00958-f003:**
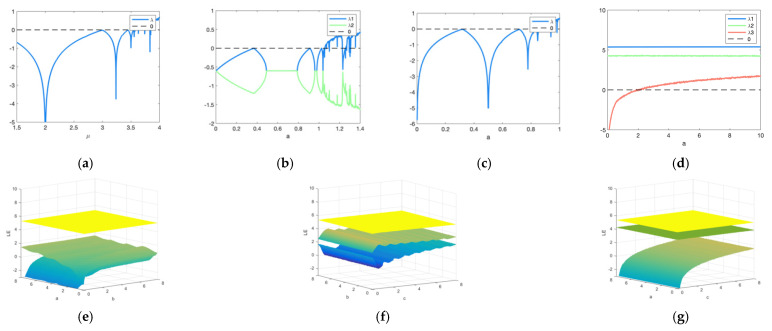
Lyapunov exponent: (**a**) Lyapunov exponent diagram of Logistic map; (**b**) Lyapunov exponent diagram of Hénon map; (**c**) Lyapunov exponent diagram of Sine map; (**d**) Lyapunov exponent diagram of Hénon map; (**e**) Lyapunov exponent diagram when a,b ∈ [0, 8], c = 0.23; (**f**) Lyapunov exponent diagram when b,c ∈ [0, 8], a = 56; (**g**) Lyapunov exponent diagram when a,c ∈ [0, 8], b = 78.

**Figure 4 entropy-24-00958-f004:**
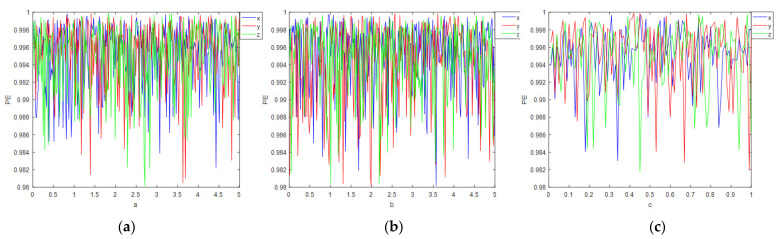
Permutation entropy controlled by a, b, c parameters: (**a**) permutation entropy when a ∈ [0, 5], b = 56, c = 0.23; (**b**) permutation entropy when b ∈ [0, 5], a = 56, c = 0.23; (**c**) permutation entropy when c ∈ [0, 1], a = 56, b = 78.

**Figure 5 entropy-24-00958-f005:**
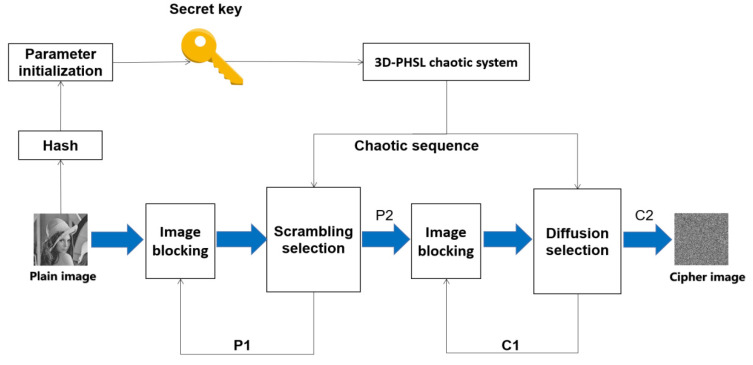
Encryption process.

**Figure 6 entropy-24-00958-f006:**
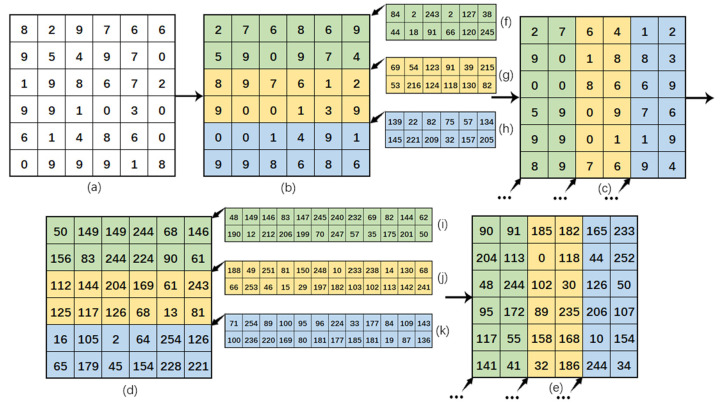
Permutation and diffusion example. (**a**) is a plaintext image; (**b**) is the result of row permutation with three permutation schemes from top to bottom; (**c**) is the result of column permutation with three permutation schemes from left to right; (**d**) is the result of row diffusion with three diffusion schemes from top to bottom; (**e**) is the result of column diffusion with three diffusion schemes from left to right; (**f**–**k**) is the adopted chaotic sequence.

**Figure 7 entropy-24-00958-f007:**
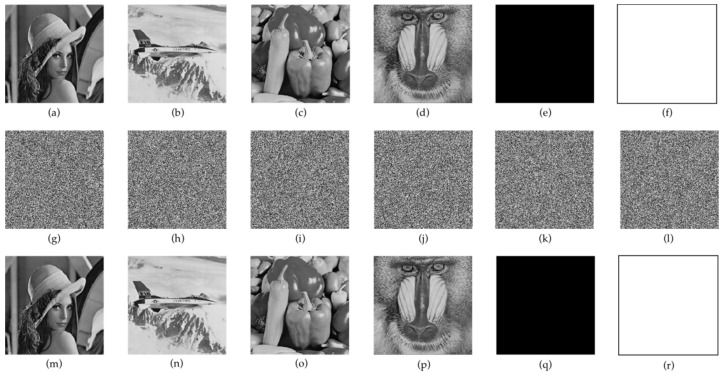
Encryption and decryption of gray-scale images. (**a**–**f**) are all kinds of plaintext images, (**g**–**l**) are corresponding encrypted images, (**m**–**r**) are corresponding decrypted images.

**Figure 8 entropy-24-00958-f008:**
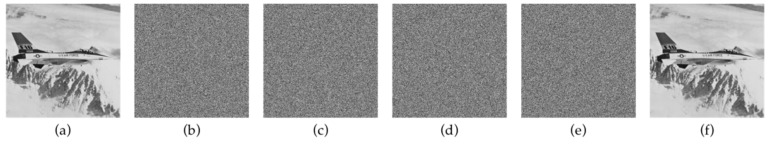
Secret key sensitivity: (**a**) plaintext Airplane; (**b**) ciphertext of Airplane; (**c**) decryption result when x = 0.69, y = 0.83, z = 0.31; (**d**) decryption result when x = 0.68, y = 0.84, z = 0.31; (**e**) decryption result when x = 0.68, y = 0.83, z = 0.32; (**f**) original initial value decryption result when x = 0.68, y = 0.83, z = 0.31.

**Figure 9 entropy-24-00958-f009:**
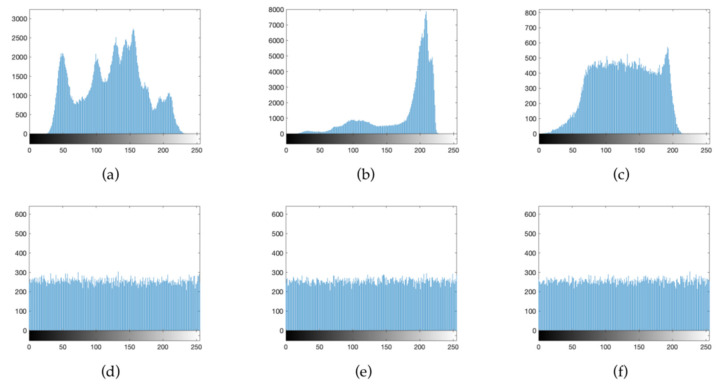
Histograms of plain images and ciphered images: (**a**) Lena plaintext histogram; (**b**) Plane plaintext histogram; (**c**) Baboon plaintext histogram; (**d**) Lena cipher histogram; (**e**) Plane cipher histogram; (**f**) Baboon cipher histogram.

**Figure 10 entropy-24-00958-f010:**
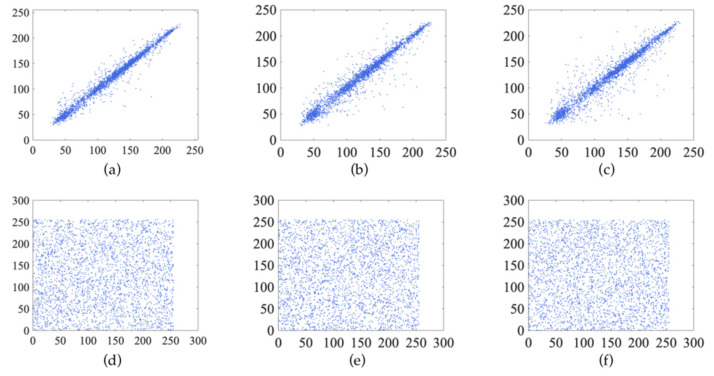
Correlation coefficients of Lena: (**a**) plaintext horizontal direction; (**b**) plaintext vertical direction; (**c**) plaintext diagonal direction; (**d**) cipher text horizontal direction; (**e**) cipher text vertical direction; (**f**) cipher text diagonal direction.

**Figure 11 entropy-24-00958-f011:**
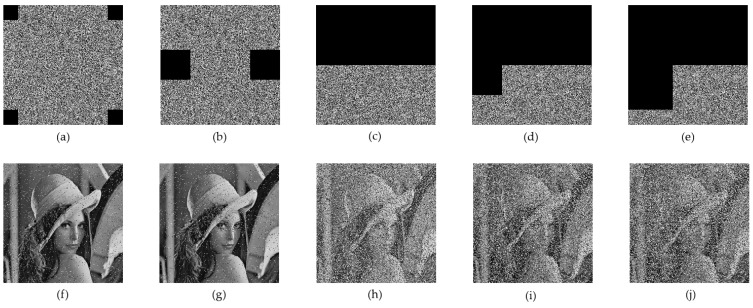
Cutting attacks of Lena: (**a**–**e**) are crop attacks at different clipping scales; (**f**–**j**) are the decryption results of (**a**–**e**).

**Figure 12 entropy-24-00958-f012:**
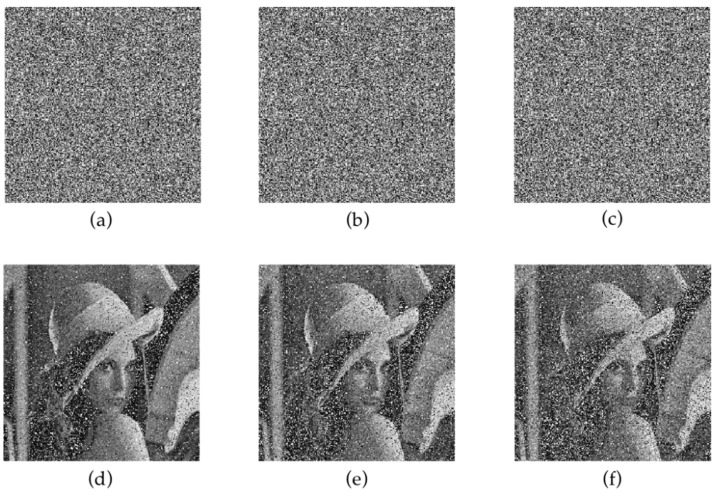
Noise attacks of Lena: (**a**) Gaussian noise attack when SNR = 25.3830; (**b**) Gaussian noise attack when SNR = 22.4203; (**c**) Gaussian noise attack when SNR = 20.7503; (**d**) decryption result of (**a**); (**e**) decryption result of (**b**); (**f**) decryption result of (**c**).

**Figure 13 entropy-24-00958-f013:**
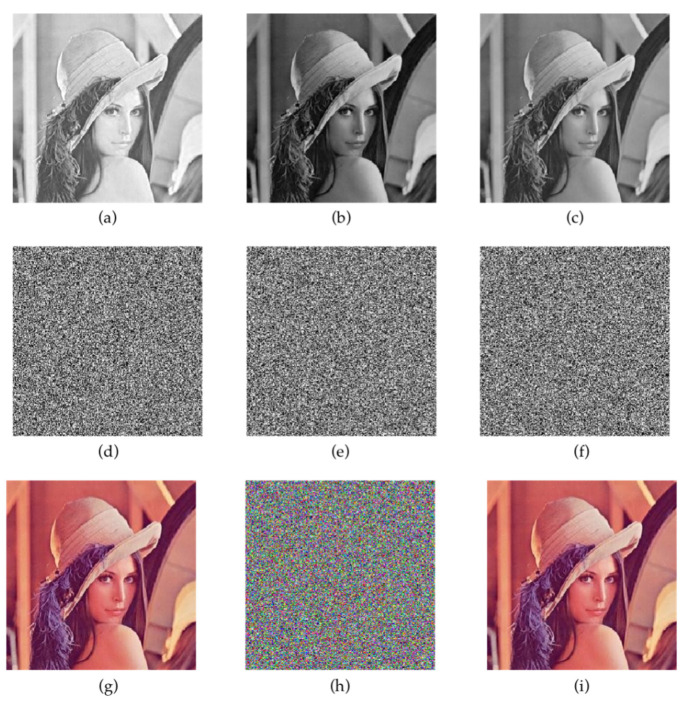
Color image encryption and decryption results: (**a**–**c**) are the R, G, B components; (**d**–**f**) are cipher tests of R, G, B component; (**g**) colored Lena; (**h**) cipher test of (**g**); (**i**) decryption result of (**g**).

**Figure 14 entropy-24-00958-f014:**
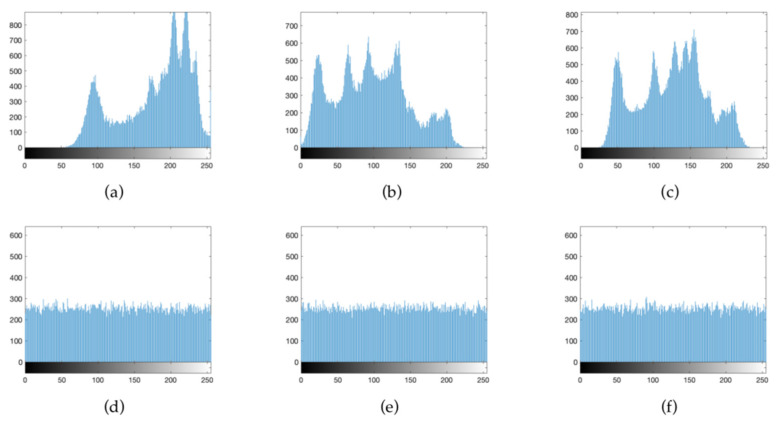
Histogram of each component before and after color image encryption: (**a**) horizontal of R component; (**b**) horizontal of G component; (**c**) horizontal of B component; (**d**) horizontal of R component; (**e**) horizontal of G component; (**f**) horizontal of B component.

**Figure 15 entropy-24-00958-f015:**
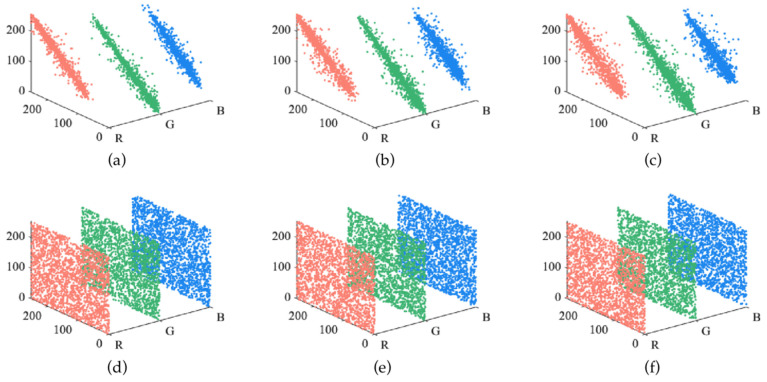
Correlation of color images before and after encryption: (**a**–**c**) are distribution of R G B component; (**d**–**f**) are distribution of encrypted R G B component.

**Figure 16 entropy-24-00958-f016:**
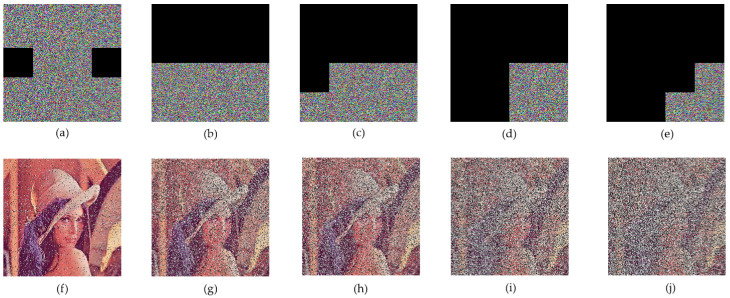
Cropping and the corresponding decryption images: (**a**–**e**) are crop attacks at different clipping scales; (**f**–**j**) are decryption results of (**a**–**e**).

**Table 1 entropy-24-00958-t001:** PE values of the different chaotic systems.

System	Parameter	PE
PHSL	a = 4.99 b = 57 c = 0.23	0.995
Logistic [21]	μ = 4	0.679
Sine [22]	μ = 1	0.669
LT [36]	μ = 4 a = 2	0.943
ICMIC [37]	c = 3	0.942
3D Logistic [38]	Γ = 3.8 β = 0.021 α = 0.014	0.987
3D Hénon [39]	m = 1.7 p = 1.0 q = 0.4	0.989

**Table 2 entropy-24-00958-t002:** Success rate of NIST test results.

The Items Used for Testing	Success Rate	The Items Used for Testing	Success Rate	The Items Used for Testing	Success Rate
**Approximate Entropy**	100%	**Linear Complexity**	95%	**Random Excursions Variant**	99.5%
**Block Frequency**	98.5%	**Longest Run**	96.5%	**Rank**	99%
**CumulativeSums**	96.5%	**NonOverlapping Template**	98.5%	**Runs**	99%
**FFT**	99.5%	**Overlapping Template**	98.5%	**Serial**	98%
**Frequency**	95.5%	**Random Excursions**	99%	**Universal**	100%

**Table 3 entropy-24-00958-t003:** Correlation coefficients of images.

Image		Plain	Proposed	Ref. [8]	Ref. [30]	Ref. [38]	Ref. [40]	Ref. [41]	Ref. [42]
Lena	Horizontal	0.93853	−0.0018	0.0061	0.0083	0.0054	0.0023	—	0.0056
	Vertical	0.9702	−0.0017	0.0116	−0.0021	0.0063	0.0019	—	0.0037
	Diagonal	0.91697	0.0001	0.0018	−0.0025	0.0023	0.0011	—	0.0032
Plane	Horizontal	0.94437	0.0003	0.0054	−0.0209	—	0.0062	0.0012	0.0028
	Vertical	0.93332	−0.0064	0.0089	0.0083	—	0.0074	−0.0063	0.0041
	Diagonal	0.89198	−0.0033	0.0021	−0.0070	—	0.0009	0.0058	0.0010
Pepper	Horizontal	0.96038	0.0039	0.0049	0.0067	—	0.0037	0.0001	0.0016
	Vertical	0.97153	−0.0026	0.0031	−0.0050	—	0.0258	−0.0008	0.0059
	Diagonal	0.93645	−0.0034	0.0079	−0.0059	—	0.0079	0.0002	0.0034
Baboon	Horizontal	0.86456	−0.0066	0.0060	—	—	0.0059	—	0.0026
	Vertical	0.82162	0.0007	0.0058	—	—	0.0041	—	0.0009
	Diagonal	0.77757	−0.0024	0.0016	—	—	0.0028	—	0.0052

**Table 4 entropy-24-00958-t004:** NPCR of different images encrypted by different schemes.

Image	Proposed	Ref. [8]	Ref. [38]	Ref. [40]	Ref. [41]	Ref. [42]
Lena	0.996048	0.996152	0.9961	0.996304	—	0.996002
Plane	0.996025	0.994350	—	0.994883	0.9967	0.996261
Pepper	0.996071	0.996202	—	0.993017	0.9970	0.996112
Cameraman	0.996084	0.996405	—	0.992052	—	0.996082
Baboon	0.996155	0.995966	—	0.992394	—	0.995903
Average value	0.996077	0.995815	—	0.99373	0.9969	0.996072

**Table 5 entropy-24-00958-t005:** UACI of different images encrypted by different schemes.

Image	Proposed	Ref. [8]	Ref. [38]	Ref. [40]	Ref. [41]	Ref. [42]
Lena	0.334389	0.335024	0.3343	0.335989	—	0.335079
Plane	0.334199	0.334109	—	0.333562	0.3361	0.335782
Pepper	0.335157	0.335323	—	0.330026	0.3358	0.335265
Cameraman	0.333959	0.334109	—	0.334390	—	0.335574
Baboon	0.335558	0.335016	—	0.333144	—	0.335281
Average value	0.3346524	0.3347162	—	0.3334222	0.3360	0.3353962

**Table 6 entropy-24-00958-t006:** Information entropy of images.

Image	Plain	Proposed	Ref. [8]	Ref. [23]	Ref. [38]	Ref. [40]	Ref. [41]	Ref. [42]
512 × 512								
Cameraman	7.0480	7.9993	7.9993	7.9923	-	7.9972	-	7.9993
Lena	7.4451	7.9992	7.9995	7.9924	7.9974	7.9994	-	7.9994
Plane	6.7135	7.9993	7.9991	7.9925	-	7.9991	7.9990	7.9992
Pepper	6.7624	7.9993	7.9990	7.9921	-	7.9983	-	7.9993
Baboon	7.2925	7.9992	7.9990	7.9922	-	7.9981	7.9989	7.9992

**Table 7 entropy-24-00958-t007:** *χ*^2^ test.

Image	Lena	Plane	Pepper	Baboon	Black	White
Plain	158,345	17,446	31,989	42,256	16,711,680	16,711,680
Proposed	255.8	265.1	250.32	252.3	261.7	264.9

**Table 8 entropy-24-00958-t008:** Encryption and decryption time.

	Encryption Time (s)	Decryption Time (s)
Ours (256 × 256)	0.176182	0.189482
Ours (512 × 512)	0.526771	0.514788

**Table 9 entropy-24-00958-t009:** Time complexity.

Proposed	Ref. [51]	Ref. [49]	Ref. [50]
O(m^2^ + n^2^ + m + n)	$O(108\mathrm{MN}+72L^{4}$)	O(MNlog(MN) + 4MNlog(4MN)) + 3O(4MN)	O(M^2*N^2)

**Table 10 entropy-24-00958-t010:** R, G, B correlation coefficients of Lena color image.

Plain Image	Horizontal	Vertical	Diagonal
R	0.9411	0.9705	0.9164
G	0.9434	0.9719	0.9204
B	0.88989	0.9433	0.8493
Cipher image			
R	−0.0035	0.0010	0.0008
G	0.0006	−0.0062	0.0033
B	0.0006	−0.0068	0.0023

## Data Availability

Not applicable.

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
