# Peer review of "Image Security Based on Three-Dimensional Chaotic System and Random Dynamic Selection"

_entropy, 2022, doi:10.3390/e24070958_

Round 1
Reviewer 1 Report
I have several comments as follows.
1- The proposed algorithm should be compared with 3D chaotic maps (Tables 1, 2, 3,4, 5 and 7).
2- The time complicity is very large O(m^3 n + n^3 m). So, how is the encryption time of the proposed algorithm less than the other algorithms?
3- The time complicity should be compared with the other algorithms.
4- All equations should have the same alignment.
5- The NIST tests should be performed to check the randomness of the generated sequences.
6- How are some figures displayed on one page and their captions on another page?
7- Try to enhance the writing of the permutation algorithms in pages 6, 7 and 8.
8- (9) and (10) are not equations. They are codes.
9- The device specifications and the application (e.g., Matlab) should be mentioned.
10- Many typos are displayed in this paper:
For example:
In abstract: line 5: change " -----, in the first step, The chaotic system ------" to " -----, in the first step, the chaotic system ------.
line 8: change 9n to 9^n
In Section 3: line 2: change "selectthe" to "select the"
In Step 5, page 8, line 11: change "algorithmbased" to "algorithm based"
and etc.
11- Correct equation (14).
Reviewer 2 Report
In the study, a novel image encryption algorithm and analysis based on a chaotic system are presented. The introduced novel image encryption algorithm is important for secure image encryption.
It would be better if the you consider some of the points I mentioned below:
1- At the end of paragraph 3 in the Introduction section, "Classic chaotic systems are Logistic[21] Sine[22] and Tent[23], etc." has an expression. The examples (Logistic Sine and Tent) mentioned in the sentence are chaotic maps. What is meant by the definition of "classical chaotic systems"?
2- As it is stated in the sentence of "Hua et al. [25] proposed an image scheme with 2D Hemon chaotic systems" in the introduction section, the study was based on sine chaotic map rather than Henon chaotic system as it is stated in the sentence. This mistake must be corrected.
3- "Hemon" is written instead of "Hènon" in some places throughout the work. These mistakes must be corrected.
4- In Section-2.24, information should be given about what "Permutation Entropy" is and how it is calculated in the study.
5- Information about the "Secret Key" used in the encryption algorithm designed in Chapter-3 should be given. What is the structure of "Secret Key"? It should be explained in detail.
6- In order to better understand the steps (Step1, Step2, etc.) of each encryption algorithm described in Chapter-3, the symbols (H, O, S, K, etc.) used should be given more clearly.
7- No information is given about the decryption process required for the encryption algorithm presented in the article. Information should also be given about the decryption design.
8- The performance and security analyzes of the study are always given over "Lena, plane, pepper, baboon, black and white" images. However, in Section-4.2.1, analyzes are given over the "Lake" picture. Instead, it would be better to use one of the "Lena, plane, pepper, baboon, black and white" images referenced in the study for analysis.
9- In Section 4.2.4, the ideal values of NPCR and UACI values should also be given in the text.
10- More detailed values should be given in Section-4.2.6 indicating the noise signal attack test, such as the type of noise signal used for noise attack testing and "noise signal power/original signal power" in dB.
11- Encryption speed is mentioned in Section-4.2.8 and compared with a few other studies. However, it is not mentioned under which electronic hardware features (PC with which feature or microcontroller with which feature, etc.) are calculated at these values. Also, do other studies have the same hardware features?
12- In Section-4.2.9, it is not fully understood where the specified 2^32 value (for each of the system initial values and system parameters) is found?
13- In Figure-13-h, the result of the decrypted color image is seen as an encrypted image!!!
14- Incorrect punctuation marks and English expression in some parts of the work should be corrected.
Round 2
Reviewer 1 Report
Correct the second part(UACI) of equation (10). All other comments are addressed.
Reviewer 2 Report
Thanks for addressing all my concerns.